# Prevalence of advanced hepatic fibrosis and individualization of associated risk factors by Bayesian analysis in MASLD patients in French cardio-metabolic health networks

**Michel Doffoel**[1,2☯], **Frédéric Chaffraix**[1,2,3☯]*, **Archia Chahard**[4‡], **Dominique Gras**[5],
**Odile Bonomi**[6], **Corinne Bildstein**[7], **Simona Tripon**[1,2], **Maude Royant**[1,2], **Nicolas Meyer**[4‡]

1 Association de Lutte contre les Maladies du Foie ALMAF, Strasbourg, France, 2 Service Expert de Lutte contre les Hépatites Virales d'Alsace SELHVA, Pôle Pathologies Hépatiques et Digestives, Nouvel Hôpital Civil, Hôpitaux Universitaires de Strasbourg, Strasbourg, France, 3 SOS Hépatites Alsace Lorraine, Strasbourg, 4 Département de Santé Publique Santé au Travail et Hygiène Hospitalière, Hôpitaux Universitaires de Strasbourg, Strasbourg, France, 5 Réseau Diabète Obésité Maladies Cardiovasculaires REDOM, Pole APSA, Strasbourg, France, 6 Réseau de Cardio Prévention Obésité Alsace RCPO, Pole APSA, Saint-Nabor, France, 7 Réseau Santé Colmar RSC, Pole APSA, Colmar, France

☯ These authors contributed equally to this work.
‡ These authors also contributed equally to this work.
* frederic.chaffraix@chru-strasbourg.fr

## Abstract

The aim of this study was to determine the prevalence of advanced hepatic fibrosis and to individualize using Bayesian analysis its associated risk factors in patients with metabolic dysfunction-associated steatotic liver disease (MASLD) being cared for in three Alsatian cardio-metabolic health networks in the North East of France. Overall, 712 patients aged ≥18 years with a female predominance were included into a prospective, cross-sectional, and observational study. Advanced fibrosis and severe steatosis were evaluated using transient elastography (FibroScan®). The proportion of MASLD patients was 80% and 84% in women and men (difference -4.2% [-10.0; 1.9]), respectively. Advanced fibrosis was observed in 11% of patients, being more common in men (16.9%) than women (7.5%) (difference 9.4 [4.3–15.0]). Severe steatosis was also more common in men (74.9%) than women (63.4%) (difference 11.4 [4.2–18.2]). Only three of the tested variables were likely associated with advanced fibrosis: gender (OR: 1.78 [1.17–2.68]; Pr [OR >1] = 1), T2DM (OR: 1.54 [1–2.37]; Pr [OR >1] = 0.97) and hypertriglyceridemia (OR: 1.49 [0.97–2.27]; Pr (OR >1) = 0.97). In conclusion, this study confirmed the usefulness of assessing hepatic fibrosis in patients with metabolic dysfunction. Therefore, access to FibroScan® should be facilitated in all cardio-metabolic health networks.

## Introduction

Metabolic dysfunction-associated steatotic liver disease (MASLD), formerly named non-alcoholic fatty liver disease (NAFLD), is constantly increasing [1]. In a meta-analytic study

**Data Availability Statement:** All relevant data are within the paper and its Supporting Information files.

**Funding:** The author(s) received no specific funding for this work.

**Abbreviations:** ALMAF, Association de Lutte contre les Maladies du Foie / French association for the fight against liver disease; CAP, Controlled Attenuation Parameter; CPP, Comité de Protection des Personnes / Personal Protection Committee; LSM, Liver Stiffness Measurements; MASLD, Metabolic dysfunction-Associated Steatotic Liver Disease; NAFLD, Non-Alcoholic Fatty Liver Disease; RCPO, Réseau de Cardio Prévention Obésité Alsace/ Alsace Cardio Prevention Obesity Alsace; REDOM, REDOM (Réseau Diabète Obésité Maladies cardiovasculaires / Diabetes Obesity Cardiovascular diseases Network; RSC, Réseau Santé Colmar/ Colmar Health Network; SELHVA, Service Expert de Lutte contre les Hépatites Virales d'Alsace/ specialty care center for the fight against viral hepatitis in Alsace; T2DM, Type 2 Diabetes Mellitus; VCTE, Vibration Controlled Transient Elastography.

involving over 1.2 million people, Le *et al* [2] estimated an NAFLD incidence rate of 4,613 new cases per 100,000 person-years. In France, the prevalence of NAFLD is estimated at 18.2% in the general population [3]. The severity of NAFLD has been linked with the progression of hepatic fibrosis [4]. Patients with NAFLD or MASLD were reported to display a similar risk of advanced liver fibrosis in the general population of the United States of America (USA) [5]. Transient elastography is a validated imaging technique for the evaluation of hepatic fibrosis [6], as it was proven to be very effective for diagnosing advanced fibrosis F3 (bridging fibrosis) —F4 (cirrhosis) [7]. Moreover, transient elastography allows for quantifying hepatic steatosis [6]. Among the independent risk factors associated with advanced fibrosis in patients with NAFLD, the components of metabolic syndrome, especially Type 2 diabetes mellitus (T2DM) [3,8–11], gender [3,12], age [3,12,13], smoking [3,13], moderate alcohol intake [14], and elevated transaminase levels [3,15], were previously reported to be associated. The potential role of lifestyle modification in the progression of fibrosis was less well studied outside of the NAFLD treatment setting [16–18]. Likewise, NAFLD constitutes an independent risk factor for cardiovascular diseases [10]. In France, the Grand Est is one of the most affected regions with respect to obesity, T2DM, and cardiovascular diseases [19]. In the Alsace territory, several cardio-metabolic health networks actively participate to the prevention and management of all these diseases involving outpatients within multidisciplinary teams bringing together private and hospital general practitioners and specialists, nurses, social workers, dietitians, and adapted physical activity educators. The present study sought to determine the prevalence of advanced hepatic fibrosis evaluated by transient elastography in MASLD patients cared for within cardio-metabolic health networks, and to individualize its associated risk factors by Bayesian logistic regression.

## Materials and methods

This cross-sectional, observational study based on prospective subject inclusion was conducted from October 2020 to June 2022. Inclusions were delayed due to the occurrence of the Covid health crisis at the beginning of 2020.

The study was supported by the French association for the fight against liver disease ALMAF (*Association de Lutte contre les Maladies du Foie*) and specialty care center for the fight against viral hepatitis in Alsace SELHVA (*Service Expert de Lutte contre les Hépatites Virales d'Alsace*). All data were anonymized and analyzed by the Clinical Research Methodology Unit of the Public Health Department of the Strasbourg University Hospital.

The study protocol was approved by the CPP (*Comité de Protection des Personnes* / Personal Protection Committee-Ile-de-France VI) on December 11, 2019 and modified on May 12, 2021 (ID RCB: 2019-A02273-54). All patients provided written informed consent before being included into the study.

### Patients

Adults aged ≥ 18 years old were included into three cardio-metabolic health networks from the Alsace region: REDOM (Réseau Diabète Obésité Maladies cardiovasculaires / Diabetes Obesity Cardiovascular Disease Network), RCPO (Réseau de Cardio Prévention Obésité Alsace/ Alsace Cardio Prevention Obesity Alsace), and RSC (Réseau Santé Colmar/ Colmar Health Network). Patients were not included if they displayed known positive HBs Ag or anti-HCV antibodies or excessive alcohol consumption (≥30g/day in men or ≥20g/day in women) [1,20]. Patients were also not included in FibroScan® failure (technical failure or uninterpretable results) cases.

The patient characteristics, as part of the usual assessment, were collected upon entry into the network: demographic features, including age and gender; components of metabolic syndrome, including obesity with body mass index [BMI $\geq$ 30kg/m2], Type 2 diabetes mellitus (T2DM), arterial hypertension, as well as dyslipidemia (hypercholesterolemia and hypertriglyceridemia); high cardiovascular risk according to the SCORE and Framingham scores; past regular tobacco consumption whatever the quantity; reported past regular alcohol consumption <30g/day in men and <20g/day in women [1,20]; Ricci and Gagnon physical activity self-assessment test which enables determining three profiles: inactive (<18 points), active (18–35 points), and very active (>35 points) (Montreal University of Canada, modified by Laureyns and Séné) [21].

## Transient elastography

Liver stiffness measurements (LSM) were performed within 3 months following entry into the network. Vibration controlled transient elastography (VCTE) technology was employed (FibroScan device; Echosens, Paris, France). ALMAF portable device (FibroScan 430 compact plus) was applied, enabling the examination to be carried out in the premises of the different networks. Examinations were considered reliable only if $\geq$10 LSMs were obtained after a fasting time $\geq$3 hours (h), with an interquartile range/median <30 LSMs. The M-probe or the XL-probe was used.

The VCTE results were expressed in kPa as the median of the valid measurements of fibrosis. The cutoff value retained for advanced fibrosis F3-F4 was 9.6 kPa whatever the probe type [22,23].

Steatosis was quantified by measuring the controlled attenuation parameter (CAP). The results were expressed in dB/m as the median of the valid measurements. The cut-off value associated with steatosis ($\geq$ S1) was 248 dB/m [24]. For severe steatosis (S3), the retained cut-off was 280 dB/m [24].

## Diagnostic criteria of NAFLD and MASLD

For NAFLD, steatosis was detected on FibroScan® based on a CAP $\geq$248dB/m, with other known causes of liver disease being excluded [25]. For MASLD, an additional metabolic syndrome component, in particular obesity or T2DM, was associated [1,25].

## Statistical analysis

Statistical analyses were done with Bayesian methods. Qualitative data were described using frequency and proportion of each modality. Quantitative data were expressed using mean (SD) or median (IQR), as well as minimum and maximum.

Description and comparison of proportions among gender and between MAFLD+ and MAFLD- were conducted using Beta uniform distribution (Beta [1,1]) in each group. For age and BMI, lowly informative Gaussian distributions were employed. The prior specified that mean age was between 20 and 100, and mean BMI between 25 and 35. Comparisons were expressed as proportion difference or as mean difference with 95% posterior credibility interval.

The role of potential risk markers on the primary outcome (advanced fibrosis) was studied using Bayesian logistic regression [26–28]. Results were expressed as odds-ratio (OR) with their 95% posterior credibility interval (95% CrI). Several assumptions were previously made on the prior distribution of OR in a sensitivity analysis run on three sets of univariate and multivariate logistic regressions. We assumed that for each variable (all binary, possibly after dichotomization for quantitative variables), the OR displayed a 95% prior probability of being

within either one of the three following intervals (1) [0.5–2]; (2) [0.5–4]; (3) [1–3], indicating a more or less strong prior knowledge on the relationship between a given potential risk marker and the outcome. These prior distributions were defined ahead of the analysis, based on the available literature and plausible physio-pathological mechanisms.

In each case, we computed the probability that the OR was larger than 1 (Pr(OR > 1 | data)) and larger than 1.5 (Pr [OR > 1.5 | data]) considering that the latter would represent a substantial effect. Missing data were rare. They have been multiply imputed under a missing completely at random (MCAR) mechanism assumption using a Bernoulli distribution whose parameter was based on the observed frequency of the predicting variable's modalities among the non-missing data.

The McMC models were run with 100 000 iterations, a burn-in of 5000 iterations, and thinning of two on a single chain. Convergence was assessed and observed in each model, using the Brooks-Gelman-Rubin test and graphical diagnostics. All computations were carried out using JAGS and R 4.2.2 through R-Studio.

## Results and discussion

In total, 744 patients were included into the three networks. Technical failure of the FibroScan® was observed in 16 patients (2.1%), with examinations being unreliable in 16 other patients (2.2%). Thus, the analyses were carried out on 712 patients, involving 280, 283, and 149 patients from the REDOM, RCPO and RSC networks, respectively.

All patient characteristics according to gender have been presented in Table 1. In the three networks, there was a female predominance, consisting of 70%, 64%, and 69% in the REDOM, RCPO and RSC networks, respectively. There was a similar proportion of MASLD in women and men, consisting of 80% and 84%, respectively (difference -4.2% [-10.0; 1.9]). Among the metabolic syndrome components, obesity and T2DM were the most common in both genders. High cardiovascular risk was more common in men (high SCORE proportion difference -30.8% [-38.2; -23.3]), as was high Framingham score (-31.7% [-38.8; -24.4] in favor of women), and past regular alcohol consumption (-31.2% [-38.4; -23.7], in favor of women). Smoking was rare in both genders. Physical inactivity concerned one-third of patients. For FibroScan®, the M probe was used in 293 patients (41.1%), and the XL probe in 417 patients (58.5%). Advanced fibrosis was observed in 11% of patients, being more common in men (16.9%) than women (7.5%) (risk difference 9.4 [4.3–15.0]). Severe steatosis was similarly more common in men (75%) than women (63%) (11.4 [4.2–18.2]). There was no center effect between the three networks for the patient characteristics and the FibroScan® data characteristics.

Patient characteristics based on MASLD presence have been presented in Table 2. The proportion of patients with MASLD represented 81%, 79%, and 85% in the REDOM, RCPO and RSC networks, respectively. The metabolic syndrome components were more common in patients with MASLD than in those without, consisting of 88% vs. 70% (difference 18.3 [10.4; 26.8]) for obesity, 38% vs. 24% (difference 13.6 [5.1; 21.5]) for T2DM, 53% vs. 41% for arterial hypertension (11.9% [2.6; 20.9]), 44% vs. 37% for hypercholesterolemia (6.9 [-2.4; 15.6]), and 31% vs. 11% for hypertriglyceridemia (19.8 [12.7; 26.0]. Physical inactivity was also more common, 49% vs. 20% (20.1 [11.9; 27.6]). On the other hand, past tobacco (2.2 [-3.2; 8.8]) and alcohol consumption (-1.4 [-10.5; 7.9]) and high cardio-vascular risk scores (high SCORE -2.4 [-11.0; 6.6] and high Framingham score -5.1 [-12.6; 3.1]) were similar in patients with or without MASLD. Advanced fibrosis was only observed in patients with MASLD (13%) (Fig 1). In this group, severe steatosis was present in 83% of patients versus 0.7% in those without MASLD (Fig 1).

**Table 1. Patient characteristics according to gender.**

| Variables | | | Female gender | Male gender | Total | Difference |
|---|---|---|---|---|---|---|
| | | | N = 481 | N = 231 | N = 712 | |
| MASLD | Yes, n (%) | | 383 (80%) | 194 (84%) | 577 (81%) | -4.2 [-10.0; 1.9] |
| | No, n (%) | | 98 (20%) | 37 (16%) | 135 (19%) | |
| Obesity BMI $\geq$ 30 kg/m2 | Yes, n (%) | | 415 (87%) | 184 (80%) | 599 (84%) | 7.0 [1.2; 13.2] |
| | No, n (%) | | 64 (13%) | 47 (20%) | 111 (16%) | |
| | Missing | | 2 | 0 | 2 | |
| | Mean (SD) | | 35.5 (5.9) | 34.6 (6.1) | 35.2 (6.0) | 1.0 [0; 1.9] |
| | Median | | 34.6 | 33.9 | 34.4 | |
| | Minimum—Maximum | | 21.4–59.8 | 16.8–56.2 | 16.8–59.8 | |
| | 1st quartile–3rd quartile | | 31.6–38.3 | 30.5–38.4 | 31.2–38.3 | |
| Type 2 diabetes mellitus | Yes, n (%) | | 151 (31%) | 103 (45%) | 254 (36%) | -13.2 [-20.7; -5.6] |
| | No, n (%) | | 330 (69%) | 128 (55%) | 458 (64%) | |
| Arterial hypertension | Yes, n (%) | | 230 (48%) | 134 (58%) | 364 (51%) | -10.4 [-18.1; -2.6] |
| | No, n (%) | | 251 (52%) | 96 (42%) | 347 (49%) | |
| | Missing | | 0 | 1 | 1 | |
| Hypercholesterolemia | Yes, n (%) | | 180 (37%) | 124 (54%) | 304 (43%) | -16.2 [-23.8; -8.5] |
| | No, n (%) | | 301 (63%) | 107 (46%) | 408 (57%) | |
| Hypertriglyceridemia | Yes, n (%) | | 112 (23%) | 84 (36%) | 196 (28%) | -12.9 [-20.2; -5.8] |
| | no, n (%) | | 365 (77%) | 147 (64%) | 512 (72%) | |
| | Missing | | 4 | 0 | 4 | |
| High cardiovascular risk | SCORE score | Yes, n (%) | 122 (25%) | 130 (56%) | 252 (35%) | -30.8 [-38.2; -23.3] |
| | | No, n (%) | 359 (75%) | 101 (44%) | 460 (65%) | |
| | Framingham score | Yes, n (%) | 78 (16%) | 111 (48%) | 189 (27%) | -31.7 [-38.8; -24.4] |
| | | No, n (%) | 403 (84%) | 120 (52%) | 523 (73%) | |
| Past regular tobacco consumption | Yes, n (%) | | 47 (10%) | 26 (11%) | 73 (10%) | -1.6 [-6.7; 3.1] |
| | No, n (%) | | 434 (90%) | 205 (89%) | 639 (90%) | |
| Past regular alcohol consumption ([a]) | Yes, n (%) | | 174 (36%) | 156 (68%) | 330 (46%) | -31.2 [-38.4; -23.7] |
| | No, n (%) | | 307 (64%) | 75 (32%) | 382 (54%) | |
| Physical activity ([b]) | Yes, n (%) | | 292 (61%) | 158 (69%) | 450 (63%) | -8.1 [-15.3; -0.6] |
| | No, n (%) | | 188 (39%) | 71 (31%) | 259 (37%) | |
| | Missing | | 1 | 2 | 3 | |

[a] < 30 g/day in men and < 20g/day in women.

[b] Ricci and Gagnon physical activity self-assessement test.

Comparisons are expressed as proportion difference and as mean difference with 95% posterior credibility interval.

Using Bayesian logistic regression, only three variables were likely associated with advanced fibrosis (Table 3), including gender (OR: 1.78 [1.17–2.68], Pr (OR >1) = 1, T2DM (OR: 1.54 [1–2.37], Pr (OR >1) = 0.97, and hypertriglyceridemia (OR: 1.49 [0.97–2.27], Pr (OR >1) = 0.97. For each of these variables, the probability of a substantial increase in OR (Pr [OR >1.5]) was less than 0.80. In a sensibility analysis, a prior giving a slight probability that OR could be larger than 1 (second prior OR between 0.5 and 4) led to no substantial modification of the conclusion. In a third analysis with a prior OR between 1 and 3, the probability of an OR larger than 1 was similar for gender, T2DM, and hypertriglyceridemia (p>0.99 or p = 1), whereas the probability of a substantial OR (OR >1.5) was larger only for women as compared with men (PrOR gender >1.5) = 0.93. In addition, several other variables displayed an increased OR, including age, obesity, hypertension, and severe steatosis, but none with a substantial increase

**Table 2. Comparison of characteristics between MASLD- and MASLD+ patients.**

| Variables | | | MASLD - | MASLD + | Difference |
|---|---|---|---|---|---|
| | | | N = 135 | N = 577 | |
| Gender | Female | | 98 (73%) | 383 (66%) | 6.0 [-2.7; 14.1] |
| | Male | | 37 (27%) | 194 (34%) | |
| Obesity BMI ≥ 30 kg/m2 | Yes, n (%) | | 94 (70%) | 505 (88%) | -18.3 [-26.8; -10.4] |
| | No, n (%) | | 41 (30%) | 70 (12%) | |
| | Missing | | 0 | 2 | |
| | Mean (SD) | | 32.0 (5.4) | 36.0 (5.9) | -4,0 [-5,0; -3,0] |
| | Median | | 31.5 | 35.3 | |
| | Minimum—Maximum | | 16.8–54.1 | 21.4–59.08 | |
| | 1st quartile–3rd quartile | | 29.4–34.1 | 32.0–39.8 | |
| Type 2 diabetes mellitus | Yes, n (%) | | 33 (24%) | 221 (38%) | -13.6 [-21.5; -5.1] |
| | No, n (%) | | 102 (76%) | 356 (62%) | |
| Arterial hypertension | Yes, n (%) | | 56 (41%) | 308 (53%) | |
| | No, n (%) | | 79 (59%) | 268 (47%) | -11.9 [-20.9; -2.6] |
| | Missing | | 0 | 1 | |
| Hypercholesterolemia | Yes, n (%) | | 50 (37%) | 254 (44%) | -6.9 [-15.6; 2.4] |
| | No, n (%) | | 85 (63%) | 323 (56%) | |
| Hypertriglyceridemia | Yes, n (%) | | 15 (11%) | 181 (31%) | -19.8 [-26.0; -12.7] |
| | No, n (%) | | 118 (89%) | 394 (69%) | |
| | Missing | | 2 | 2 | |
| High cardiovascular risk | SCORE score | Yes, n (%) | 45 (33%) | 207 (36%) | -2.4 [-11.0; 6.6] |
| | | No, n (%) | 90 (67%) | 370 (64%) | |
| | Framingham score | Yes, n (%) | 30 (22%) | 159 (28%) | -5.1 [-12.6; 3.1] |
| | | No, n (%) | 105 (78%) | 418 (72%) | |
| Past regular tobacco consumption | Yes, n (%) | | 16 (12%) | 57 (10%) | 2.2 [-3.2; 8.8] |
| | No, n (%) | | 119 (88%) | 520 (90%) | |
| Past regular alcohol consumption ([a]) | Yes, n (%) | | 61 (45%) | 269 (47%) | -1.4 [-10.5; 7.9] |
| | No, n (%) | | 74 (55%) | 308 (53%) | |
| Physical activity ([b]) | Yes, n (%) | | 108 (80%) | 342 (60%) | 20.1 [11.9; 27.6] |
| | No, n (%) | | 27 (20%) | 232 (40%) | |
| | Missing | | 0 | 3 | |

[a] < 30 g/day in men and < 20g/day in women.

[b] Ricci and Gagnon physical activity self-assessement test.

Comparisons are expressed as proportion difference and as mean difference with 95% posterior credibility interval.

in OR. Past regular alcohol consumption showed a moderately increased OR, but no clear association with advanced fibrosis. For past regular tobacco consumption no relationship could be established with advanced fibrosis.

Results are expressed as odds-ratio (OR) with their 95% posterior credibility interval (95% CrI). OR have a 95% prior probability (Pr) of being within either one of the three following intervals [0.5–2](first model), [0.5–4](second model) and [1–3] (third model).(Ora = adjusted OR).

## Discussion

In this study, the prevalence of advanced fibrosis in MASLD patients was 13%. Bayesian methods made it possible to demonstrate that only three of the tested variables were likely associated with advanced fibrosis, including gender, T2DM, and hypertriglyceridemia.

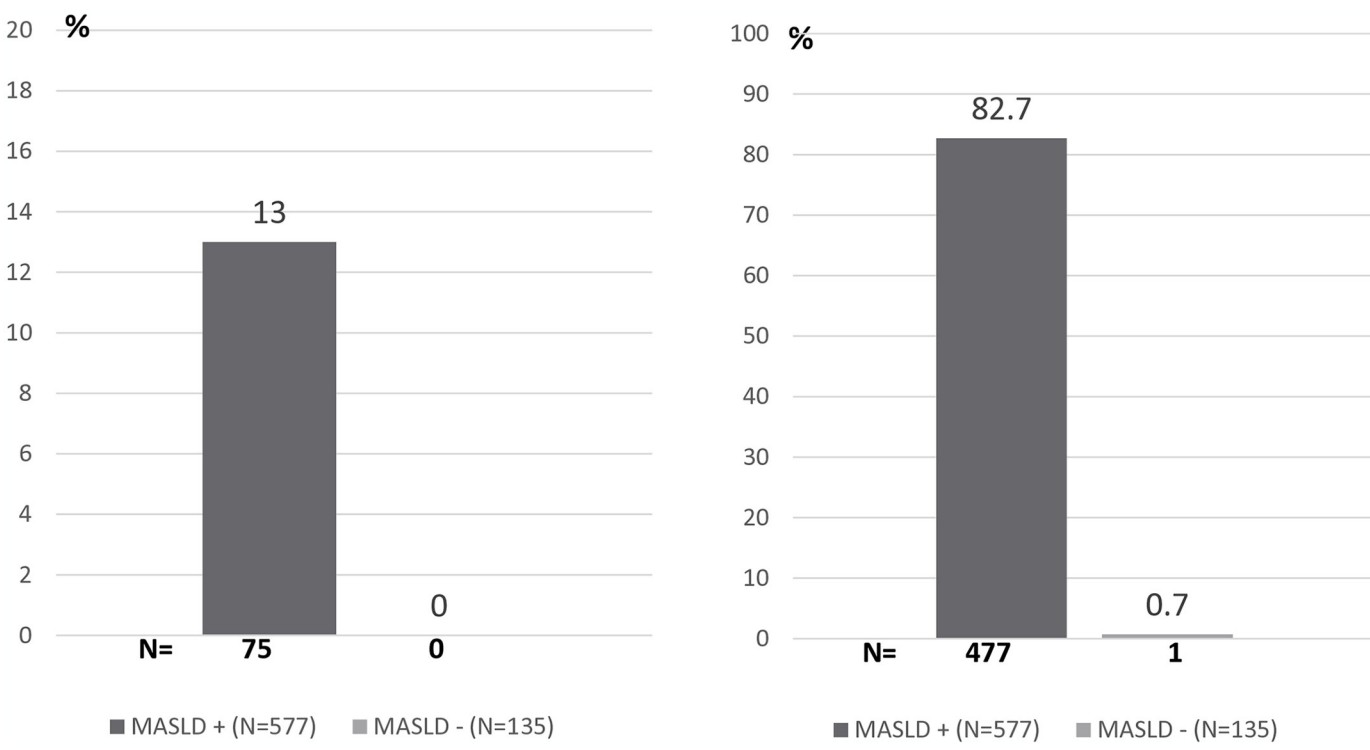

**Fig 1. Prevalence of advanced fibrosis and severe steatosis in MASLD+ and MASLD- patients.**

In NAFLD, the prevalence of advanced fibrosis assessed by liver biopsy [12,29–31] or VTCE with a cut-off of 9.6–9.7 kPa [15,32–34] was close to what we observed in our study, this prevalence being usually between 15 and 20%. The choice of VTCE as a tool for evaluating liver fibrosis has been linked to several factors. First, this choice was guided by local availability and context of use [6] in the three health networks with the logistical and qualified personal support of both ALMAF and SELHVA (hepato-gastro-enterologists; coordinating nurse). Thus, the success rate of the FibroScan was 97.9% (728/744), the examination being interpretable in 97.8% of patients (712/728), confirming the practicability of the procedure. Second, the comparison of the areas under the ROC curve AUROCs for the diagnosis of advanced fibrosis revealed VTCE to be significantly more accurate than all blood tests of fibrosis [7]. However, each modality was most reliable in excluding advanced fibrosis. Third, VTCE enabled the simultaneous quantification of fibrosis and steatosis with immediate results. Fourth, the design of our study was developed prior to the availability of diagnostic algorithms for using noninvasive tests for risk stratification of patients with suspect NAFLD in clinical practice [35–37]. The cut-off of 9.6kPa for the diagnosis of advanced fibrosis (F3-F4) was retained in our study for the following arguments. First, this was the threshold used in Hagstrom's meta-analysis [4], as well as in several studies focused on the screening of advanced fibrosis in diabetic patients [32,33,38]. Second, very close cut-offs, between 9.7 and 10 kPa, were employed in other studies concerning NAFLD or T2DM [15,23,39]. In our study, the same cut-off was retained with the M probe and XL probe, given that in the multivariable analysis of Eddowes et al, the probe type had no impact on LSM [23]. The cut-off values associated with steatosis and severe steatosis corresponded to those retained in the Karlas et al [24] meta-analysis and in the two Lee et al and Alfadda et al studies involving T2DM patients [39,40]. The cut-off

**Table 3. Results of Bayesian logistic regression.**

| Characteristics | | Advanced fibrosis No (N = 502) | Advanced fibrosis Yes (N = 75) | Prior OR [0.5–2] OR [CrI 95%] | Pr > 1 | Pr > 1.5 | Prior OR [0.5–2] Ora [CrI 95%] | Pr > 1 | Pr > 1.5 | Prior OR [0.5–4] OR [CrI 95%] | Pr > 1 | Pr > 1.5 | Prior OR [0.5–4] Ora [CrI 95%] | Pr > 1 | Pr > 1.5 | Prior OR [1–3] OR [CrI 95%] | Pr > 1 | Pr > 1.5 | Prior OR [1–3] Ora [CrI95%] | Pr > 1 | Pr > 1.5 |
|---|---|---|---|---|---|---|---|---|---|---|---|---|---|---|---|---|---|---|---|---|---|
| Age class | < 50 years | 129 (26%) | 14 (19%) | - | | | - | | | - | | | - | | | - | | | - | | |
| | ≥ 50 years | 373 (74%) | 61 (81%) | 1.27 [0.82:2.01] | 0.85 | 0.24 | 1.24 [0.78:2.01] | 0.82 | 0.22 | 1.27 [0.82:2.01] | 0.85 | 0.24 | 1.52 [0.87:2.73] | 0.85 | 0.51 | 1.27 [0.82:2.01] | 0.85 | 0.24 | 1.55 [1.01:2.4] | 0.98 | 0.56 |
| Gender | Female | 347 (69%) | 36 (48%) | - | | | - | | | - | | | - | | | - | | | - | | |
| | Male | 155 (31%) | 39 (52%) | 1.81 [1.21:2.69] | 1 | 0.81 | 1.78 [1.17:2.68] | 1 | 0.79 | 1.81 [1.21:2.89] | 1 | 0.81 | 2.24 [1.39:3.6] | 1 | 0.95 | 1.81 [1.21:2.69] | 1 | 0.81 | 2 [1.37:2.91] | 1 | 0.93 |
| Obesity BMI ≥ 30 kg/m2 | No | 64 (13%) | 6 (8%) | - | | | - | | | - | | | - | | | - | | | - | | |
| | Yes | 436 (87%) | 69 (92%) | 1.25 [0.75:2.12] | 0.8 | 0.25 | 1.28 [0.76:2.2] | 0.82 | 0.28 | 1.25 [0.75:2.12] | 0.8 | 0.25 | 1.66 [0.85:3.36] | 0.93 | 0.61 | 1.25 [0.75:2.12] | 0.8 | 0.25 | 1.81 [1.15:2.9] | 0.99 | 0.79 |
| Type 2 diabetes mellitus | No | 322 (64%) | 34 (45%) | - | | | - | | | - | | | - | | | - | | | - | | |
| | Yes | 180 (36%) | 41 (55%) | 1.67 [1.12:2.49] | 0.99 | 0.71 | 1.54 [1:2.37] | 0.97 | 0.55 | 1.67 [1.13:2.49] | 0.99 | 0.71 | 1.8 [1.09:2.97] | 0.99 | 0.76 | 1.67 [1.12:2.49] | 0.99 | 0.71 | 1.65 [1.11:2.43] | 0.99 | 0.68 |
| Arterial hypertension | No | 240 (48%) | 28 (37%) | - | | | - | | | - | | | - | | | - | | | - | | |
| | Yes | 261 (52%) | 47 (63%) | 1.33 [0.89:2] | 0.92 | 0.28 | 1.23 [0.8:1.89] | 0.83 | 0.18 | 1.33 [0.89:2] | 0.92 | 0.28 | 1.37 [0.84:2.23] | 0.89 | 0.36 | 1.33 [0.89:2] | 0.92 | 0.28 | 1.41 [0.96:2.09] | 0.96 | 0.38 |
| Hypercholesterolemia | No | 285 (57%) | 38 (51%) | - | | | - | | | - | | | - | | | - | | | - | | |
| | Yes | 217 (43%) | 37 (49%) | 1.18 [0.79:1.76] | 0.79 | 0.12 | 0.94 [0.61:1.45] | 0.39 | 0.02 | 1.18 [0.79:1.76] | 0.79 | 0.12 | 0.99 [0.55:1.51] | 0.37 | 0.03 | 1.18 [0.79:1.76] | 0.79 | 0.12 | 1.12 [0.76:1.65] | 0.71 | 0.07 |
| Hypertriglyceridemia | No | 354 (71%) | 40 (53%) | - | | | - | | | - | | | - | | | - | | | - | | |
| | Yes | 146 (29%) | 35 (47%) | 1.65 [1.09:2.46] | 0.99 | 0.68 | 1.49 [0.97:2.27] | 0.97 | 0.48 | 1.65 [1.09:2.46] | 0.99 | 0.68 | 1.7 [1.03:2.78] | 0.98 | 0.69 | 1.65 [1.09:2.46] | 0.99 | 0.68 | 1.63 [1.11:2.4] | 0.89 | 0.67 |
| High cardiovascular risk — SCORE score | No | 328 (65%) | 42 (56%) | - | | | - | | | - | | | - | | | - | | | - | | |
| | Yes | 174 (35%) | 33 (44%) | 1.32 [0.86:1.93] | 0.89 | 0.24 | 1 [0.61:1.61] | 0.49 | 0.05 | 1.3 [0.86:1.93] | 0.89 | 0.24 | 0.96 [0.53:1.72] | 0.44 | 0.07 | 1.3 [0.86:1.93] | 0.89 | 0.24 | 1.1 | 0.8 | 0.18 |
| High cardiovascular risk — Framingham score | No | 368 (73%) | 50 (67%) | | | | | | | | | | | | | | | | | | |
| | Yes | 134 (27%) | 25 (33%) | | | | | | | | | | | | | | | | | | |
| Past regular tobacco consumption | No | 449 (89%) | 71 (95%) | - | | | - | | | - | | | - | | | - | | | - | | |
| | Yes | 53 (11%) | 4 (5.0%) | 0.8 [0.44:1.33] | 0.18 | 0.01 | 0.8 [0.44:1.31] | 0.17 | 0.01 | 0.83 [0.39:1.53] | 0.24 | 0.03 | 0.81 [0.38:1.47] | 0.22 | 0.02 | 1.24 [0.77:1.90] | 0.8 | 0.18 | 1.23 [0.76:1.87] | 0.79 | 0.17 |
| Past regular alcohol consumption (a) | No | 271 (54%) | 37 (49%) | - | | | - | | | - | | | - | | | - | | | - | | |
| | Yes | 231 (46%) | 38 (51%) | 1.07 [0.69:1.59] | 0.59 | 0.05 | 1.16 [0.76:1.69] | 0.73 | 0.08 | 1.13 [0.68:1.76] | 0.65 | 0.1 | 1.27 [0.80:1.93] | 0.83 | 0.2 | 1.35 [0.90:1.94] | 0.93 | 0.26 | 1.44 [0.98:2.03] | 0.96 | 0.37 |
| Physical activity (b) | Yes | 301 (60%) | 41 (55%) | - | | | - | | | - | | | - | | | - | | | - | | |
| | No | 198 (40%) | 34 (45%) | 1.17 [0.78:1.74] | 0.78 | 0.11 | 1.19 [0.79:1.78] | 0.79 | 0.13 | 1.17 [0.78:1.74] | 0.78 | 0.11 | 1.34 [0.84:2.13] | 0.89 | 0.32 | 1.17 [0.78:1.74] | 0.79 | 0.11 | 1.5 [1.03:2.18] | 0.98 | 0.5 |

*(Continued)*

**Table 3.** (Continued)

| Characteristics | | Advanced fibrosis | | Prior OR [0.5–2] | | | | | | Prior OR [0.5–4] | | | | | | Prior OR [1–3] | | | | | |
|---|---|---|---|---|---|---|---|---|---|---|---|---|---|---|---|---|---|---|---|---|---|
| | | | | OR | | | Ora | | | OR | | | Ora | | | OR | | | Ora | | |
| | | No (N = 502) | Yes (N = 75) | [CrI 95%] | Pr > 1 | Pr > 1.5 | [CrI 95%] | Pr > 1 | Pr > 1.5 | [CrI 95%] | Pr > 1 | Pr > 1.5 | [CrI 95%] | Pr > 1 | Pr > 1.5 | [CrI 95%] | Pr > 1 | Pr > 1.5 | [CrI95%] | Pr > 1 | Pr > 1.5 |
| Severe steatosis | No | 94 (19%) | 6 (8%) | - | | | - | | | - | | | - | | | - | | | - | | |
| | Yes | 408 (92%) | 69 (92%) | 1.54 [0.95: 2.56] | 0.96 | 0.54 | 1.34 | 0.87 | 0.34 | 1.54 [0.95: 2.56] | 0.96 | 0.54 | 1.61 [0.84: 3.26] | 0.92 | 0.58 | 1.54 [0.95: 2.56] | 0.96 | 0.54 | 1.7 [1.07: 2.74] | 0.99 | 0.7 |

[a] < 30 g/day in men and < 20g/day in women.

[b] Ricci and Gagnon physical activity self-assessement test.

associated with significant steatosis (>33% of hepatocytes) was almost always >250dB/m [6]. Using biopsy analysis as the reference standard, Eddowes *et al* [23]found that CAP identified patients with steatosis (≥5% steatosis) with an AUROC of 0.87. In the US multicenter study of Siddiqui *et al*, using the XL-probe in 393 NAFLD patients, CAP exhibited an AUROC of 0.76 for detecting steatosis >5% and 96% positive predictive value at a cut-off of 263dB/m [41]. In our study, no association was found between severe steatosis and advanced fibrosis, as in the Eddowes et al study [23].

Only three of the tested variables individualized by Bayesian analysis were likely associated with advanced fibrosis. These three variables have already been individualized using classical statistical methods. The association with gender was observed yet with an opposite direction in the scientific literature [3,12]. This was for us a reason to use prior OR distribution centered on 1 to indicate that the total amount of knowledge on this effect is roughly similar to a lack of effect. In our study, a relationship was nevertheless observed. Among the metabolic syndrome components, T2DM was the most common risk factor of advanced fibrosis reported in NAFLD [3,8–10]. Dyslipidemia was rarely reported, but also less frequently studied [8,11].

Other variables were not associated with an increased risk of advanced fibrosis, particularly the past regular alcohol and tobacco consumption. The thresholds used in our study are consistent with those used in the recent classification of different categories of steatotic liver disease [1]. They allow to eliminate a possible MASLD with moderate alcohol intake (MetALD).

We chose to use Bayesian statistical methodology on account of its ability to include prior information derived from previous studies in order to accumulate evidence for or against an effect [26–28]. The ability to compute the probability of an effect of a given size, which is not provided by the classical p-values, was herein used to compute the probability that the OR was larger than one, possibly even substantially (OR >1.5). Thus, we were able to demonstrate that, whereas an effect was likely present for three of the potential predictors tested, this effect was probably of modest importance, given that for none of the three variables in question the probability of the OR being greater than 1.5 exceeded 0.79, except for gender, with a probability of 0.93, yet only in the third optimistic type of prior in the sensitivity analysis. This sensitivity analysis also demonstrated that our sample size was probably large enough, since the results are very similar in the three series of logistic regressions across the sensitivity analysis, even though some variables were likely to exhibit a more marked effect in the third model, with a probability larger than 0.95 that the OR was >1 for most of them. Nevertheless, the effect turned out to be only modest, given that OR exhibited only a small probability of being larger than 1.5. This was true for age, obesity, arterial hypertension, and steatosis.

This study displayed several limitations that deserve to be mentioned. It was a cross-sectional study which does not allow for the progression of fibrosis over time to be assessed, whereas only correlation could be observed and quantified. We can only hypothesize hints to causation and envision future studies to further explore and confirm these results. One cannot exclude the point that probably several unmeasured confounding factors likely play a role in the pathophysiology of severe steatosis. A more detailed and prospective study would be most useful to delineate the prognostic role of the factors to be associated with severe fibrosis herein identified. Tobacco and alcohol consumptions were only declarative. Furthermore, tobacco consumption has not been quantified. However, smoking has been associated only with an increased risk of hepatocellular carcinoma in MASLD patients [42]. Alcohol consumption has not been assessed by validated instruments and/or specific biomarkers [1]. However, in the history of alcohol consumption the previous drinking behaviour seems more important than the current drinking pattern [1]. The FibroScan® alone was used for fibrosis assessment prior to the availability of new sequential combinations of non-invasive tests providing a more accurate diagnosis of advanced fibrosis in MASLD [1]. However, at the time of study design VTCE

was significantly more accurate than all blood tests of fibrosis taken in isolation for diagnosing advanced fibrosis. In lifestyle modifications, physical activity was taken into account, whereas dietary factors were not.

## Conclusions

In conclusion, this study confirmed the usefulness of assessing hepatic fibrosis in patients with metabolic dysfunction. Accessibility to FibroScan® in cardio-metabolic health networks is conditioned on the intervention of a mobile team with the necessary equipment. As part of a sequential approach the dissemination of FIB-4, which is the most widely established and available blood test, should make it possible to restrict the number of FibroScan® and to promote their accessibility. In order to guarantee the latter a network organization at the level of a health territory with a pooling of personnel and FibroScan® would be justified with coordination by reference centers. We are developing such a strategy at the level of the Alsace territory with an equipped health vehicle. In addition, Bayesian analysis confirmed the results of classic statistical methods allowing for the individualization of several common risk factors associated with advanced fibrosis, especially diabetes.

## Acknowledgments

We wish to thank the nurse Cécile Domanke from ALMAF for the data collection, the ALMAF, SELHVA, AFD (Association Française des Diabetiques–French Association of Diabetics) and SOS Hépatites Alsace Lorraine's teams, and all health professionals of RCPO, REDOM and RSC for their investment in the realization of study.

## Author Contributions

**Conceptualization:** Michel Doffoel, Frédéric Chaffraix, Dominique Gras, Odile Bonomi, Corinne Bildstein.

**Data curation:** Frédéric Chaffraix, Dominique Gras, Odile Bonomi, Corinne Bildstein.

**Formal analysis:** Michel Doffoel.

**Methodology:** Michel Doffoel, Frédéric Chaffraix.

**Project administration:** Michel Doffoel, Frédéric Chaffraix.

**Validation:** Michel Doffoel, Frédéric Chaffraix, Archia Chahard, Nicolas Meyer.

**Writing – original draft:** Michel Doffoel, Frédéric Chaffraix.

**Writing – review & editing:** Michel Doffoel, Frédéric Chaffraix, Archia Chahard, Simona Tripon, Maude Royant, Nicolas Meyer.

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
