## [Decision Letter · Decision Letter 0]

1 Oct 2024

PONE-D-24-16542Prevalence of advanced hepatic fibrosis and individualization of associated risk factors in MASLD patients in French cardio-metabolic health networksPLOS ONE

Dear Dr. CHAFFRAIX,

Thank you for submitting your manuscript to PLOS ONE. After careful consideration, we feel that it has merit but does not fully meet PLOS ONE’s publication criteria as it currently stands. Therefore, we invite you to submit a revised version of the manuscript that addresses the points raised during the review process.

We look forward to receiving your revised manuscript.

Kind regards,

Anna Di Sessa, PhD, MD

Academic Editor

PLOS ONE

**Journal Requirements:**

"none"

**Additional Editor Comments:**

All the minor points raised by reviewers need to be carefully addressed

Reviewers' comments:

Reviewer's Responses to Questions

**Comments to the Author**

1. Is the manuscript technically sound, and do the data support the conclusions?

Reviewer #1: Yes

Reviewer #2: Yes

2. Has the statistical analysis been performed appropriately and rigorously? 

Reviewer #1: Yes

Reviewer #2: Yes

3. Have the authors made all data underlying the findings in their manuscript fully available?

Reviewer #1: No

Reviewer #2: Yes

4. Is the manuscript presented in an intelligible fashion and written in standard English?

Reviewer #1: Yes

Reviewer #2: Yes

5. Review Comments to the Author

**Reviewer #1:** The study underscores the importance of evaluating hepatic fibrosis in patients with metabolic dysfunction and suggests that noninvasive diagnostic tools like FibroScan should be integrated into clinical practice for better management of these patients. Since FibroScan was the main diagnostic tool used in this study, it would be helpful to readers if authors comment on the accessibility of the FibroScan diagnostic tool. Is this equipment widely available and affordable for clinical practice, including low and middle income settings? Having used this equipment for about two years, are there maintenance limitations that limits accessibility? Are there alternative non-invasive diagnostic tools that can function efficiently like the fibroScan?

**Reviewer #2:** The percentage of past regular alcohol consumption is more predominant in male gender, the author can try to describe or analysis the result in no regular alcohol consumption population. There are significant differences among MASLD, MetALD and MAFLD. the author may try to discussion base on your recent study result.

6. PLOS authors have the option to publish the peer review history of their article (what does this mean?). If published, this will include your full peer review and any attached files.

Reviewer #1: **Yes: **Prof. Yaw Amo Wiafe

Reviewer #2: No

---

## [Author Response · Author response to Decision Letter 0]

3 Dec 2024

Response to reviewers’ comments

Reviewer #1 :

Accessibility to FibroScan in cardio-metabolic health networks is conditioned on the intervention of a mobile team with the necessary equipment. As part of a sequential approach the dissemination of FIB-4, which is the most widely established and available blood test, should make it possible to restrict the number of FibroScan and to promote their accessibility. In order to guarantee the latter a network organization at the level of a health territory with a pooling of personnel and FibroScan would be justified with coordination by reference centers. For the record, the design of our study was developed prior to the availability of diagnostic algorithms for using noninvasive tests for risk stratification of patients with suspect NAFLD (MASLD) in clinical practice. 

Reviewer #2

The alcohol consumption thresholds used in our study are consistent with those used in the recent classification of different categories of steatotic liver disease (see reference 1). They allow to eliminate a possible MetALD (MASLD with moderate alcohol intake, 30-60g/day in men and 20-50g/day in women). 

 Indeed, alcohol consumption was more common among men than among women (132/194 =68% vs 137/383 = 36%), but it did not increase the risk of advanced fibrosis in the Bayesian logistic regression (see Table 3). For information, no effect of past regular alcohol consumption was observed for all variables studied.

---

## [Editor Report · Decision Letter 1]

8 Dec 2024

Prevalence of advanced hepatic fibrosis and individualization of associated risk factors by Bayesian analysis in MASLD patients in French cardio-metabolic health networks

PONE-D-24-16542R1

Dear Dr. CHAFFRAIX,

We’re pleased to inform you that your manuscript has been judged scientifically suitable for publication and will be formally accepted for publication once it meets all outstanding technical requirements.

Kind regards,

Anna Di Sessa, PhD, MD

Academic Editor

PLOS ONE
---

## [Editor Report · Acceptance letter]

18 Dec 2024

PONE-D-24-16542R1 

PLOS ONE

Dear Dr. Chaffraix, 

I'm pleased to inform you that your manuscript has been deemed suitable for publication in PLOS ONE. Congratulations! Your manuscript is now being handed over to our production team.

Kind regards, 

on behalf of

Dr. Anna Di Sessa 

Academic Editor

PLOS ONE